# Effective Therapeutic Verification of Crocin I, Geniposide, and Gardenia (*Gardenia jasminoides* Ellis) on Type 2 Diabetes Mellitus In Vivo and In Vitro

**DOI:** 10.3390/foods12081668

**Published:** 2023-04-17

**Authors:** Haibo Zhou, Sen Zhang, Lianghua Chen, Yimei Liu, Luhong Shen, Jiuliang Zhang

**Affiliations:** 1College of Food Science and Technology, Huazhong Agricultural University, No.1, Shizishan Street, Hongshan District, Wuhan 430070, China; 2Key Laboratory of Fujian Province for Physiology and Biochemistry of Subtropical Plant, Fujian Institute of Subtropical Botany, Xiamen 361006, China; 3Key Laboratory of Environment Correlative Dietology, Ministry of Education, Wuhan 430070, China

**Keywords:** gardenia, α-glucosidase, crocin I, geniposide, type 2 diabetes mellitus

## Abstract

For many centuries, Gardenia (*Gardenia jasminoides* Ellis) was highly valued as a food homologous Chinese herbal medicine with various bioactive compounds, including crocin I and geniposide. However, the functional mechanism underlying the hypoglycemic effect of gardenia is absent in the literature. To evaluate the effect of gardenia and its different extracts on type 2 diabetes mellitus (T2DM) in in vivo and in vitro experiments, the dried gardenia powder was extracted using 60% ethanol and eluted at different ethanol concentrations to obtain the corresponding purified fragments. After that, the active chemical compositions of the different purified gardenia fragments were analyzed using HPLC. Then, the hypoglycemic effects of the different purified gardenia fragments were compared using in vitro and in vivo experiments. Finally, the different extracts were characterized using UPLC-ESI-QTOF-MS/MS and the mass spectrometric fragmentation pathway of the two main compounds, geniposide and crocin I, were identified. The experimental results indicated that the inhibitory effect of the 40% EGJ (crocin I) on the α-glucosidase was better than the 20% EGJ (geniposide) in vitro. However, the inhibitory effect of geniposide on T2DM was better than crocin I in the animal experiments. The different results in vivo and in vitro presumed potentially different mechanisms between crocin I and geniposide on T2DM. This research demonstrated that the mechanism of hypoglycemia in vivo from geniposide is not only one target of the α-glucosidase but provides the experimental background for crocin I and the geniposide deep processing and utilization.

## 1. Introduction

T2DM is a metabolic disorder disease that is closely linked to an insufficient or defective insulin secretion. The main manifestation of T2DM is postprandial hyperglycemia [1]. Currently, there are more than 400 million people with diabetes worldwide and these records will continue to increase in the future [2]. The long-term vascular complications are the leading causes of an increased need for medical care, a reduced quality of life, and an increased risk of early death [3,4]. Amidst the strategies being developed for the management of T2DM with the established therapies, the α-glucosidase inhibitors have been approved in several treatment protocols [5]. The α-glucosidase is a kind of glucosidase that has a significant ability for digestion and hydrolysis starching in the small intestine [6]. The hypoglycemic principle inhibits the starch digestion in the small intestine and delays the absorption of glucose by inhibiting the α-glucosidase, thereby preventing the rate of abnormal postprandial glucose production [7]. Although hypoglycemic drugs, such as acarbose, metformin, and thiazolidinediones, clinically restrict the glucose absorption, adverse events to the undesirable gastrointestinal symptoms impede their application [8]. As a result, natural and harmless remedies are constantly being sought, and the search for natural products to treat diabetes is a viable and valuable approach.

The typical clinical symptoms of type 2 diabetes are polydipsia, polyuria, polyphagia, and weight loss. Gardenia can quench thirst and is widely used alone or in combination with other medicines for the treatment of T2DM in China and throughout Asia (The Pharmacopoeia Commission of PRC, 2015). Gardenia is a kind of homologous food medicine that is rich in various bioactive active compounds, such as geniposide and crocetin (including crocin I and crocin II) [9]. Geniposide is the most abundant iridoid and is the material basis of its biological activity. Geniposide has a significant effect on the digestive, cardiovascular, and central nervous systems [10,11,12]. Crocetin, also known as crocin, is divided into crocin I and crocin II, according to the type of the connection disaccharides at both ends of the crocin acid [13]. Crocin I accounts for 70% of the crocetin [14]. As crocetin is rare and precious, it has become the main trend to acquire it from gardenia. In recent years, gardenia was found to have many pharmacological benefits, such as anti-inflammatory effects, tumor prevention, cardiovascular disease prevention, and water-soluble nutrients that can be easily absorbed in the body [15,16,17]. One study showed that the *gardenia latifolia* extract (GLE) had an anti-diabetic potential in rats with a high-fat diet (HFD) + streptozotocin (STZ) induced type 2 diabetes mellitus (T2DM) [18]. However, there was less information about the functional mechanism of the hypoglycemic effects of crocin I, geniposide, from gardenia on T2DM in vivo and in vitro. Therefore, it was necessary to investigate and compare the intervention of geniposide, crocin I, and the extract of gardenia on hyperglycemic mice.

The objective of the present study was to compare the hypoglycemic effects of the bioactive compounds from gardenia in in vivo and in vitro experiments and to identify their structures. First, the major bioactive components of the extract fractions were identified using analytical methods. Next the activity of the major bioactive components against the enzyme and on the mice were compared. Then, different concentrations of alcohol were used to isolate and purify the crude extracts of gardenia (EGJ). Finally, geniposide and crocin I were obtained. This paper explored the hypoglycemic role-playing effects of the geniposide, crocin I, and EGJ intervention on T2DM. This research can also enrich the potential use of gardenia as a hypoglycemic functional food and lay the foundation for a new high added value product development of gardenia.

## 2. Materials and Methods

### 2.1. Materials and Reagents

Gardenia (*G. jasminoides* Ellis) from the Hubei province in China was provided and characterized by Hubei HaoYSJ Biotechnology Co. Limited (Yichang, China). The voucher specimens of gardenia (No. 190925) were deposited in the College of Food Science and Technology, Huazhong Agricultural University. The α-glucosidase from *Saccharomyces cerevisiae* (EC 3.2.1.20) was obtained from Sigma-Aldrich (St. Louis, MO, USA). The *p*-nitrophenyl (*p*NPG), α-D-glucosidase (purity > 99%), and acarbose (purity > 95%) were purchased from Yuanye Biological Technology (Shanghai, China). The AB-8 macroporous adsorption resin (0.3–1.25 mm particle size) was purchased from Nankai Hecheng Science & Technology Co. (Tianjin, China). The crocin I and geniposide (HPLC grade) were purchased from Shanghai Yuanye Biotechnology Co. (Shanghai, China). The methanol and acetic acid (HPLC grade) were obtained from J.T. Baker Co. (Phillipsburg, NJ, USA) and Aladdin Industrial Co. (Shanghai, China), respectively. The water was purified using a Milli-Q system supplied by Millipore (Billerica, MA, USA). All the other reagents were of an analytical grade.

### 2.2. Isolation and Identification of Geniposide and Crocetin

The purified and modified fractions of gardenia were mainly prepared according to the preliminary laboratory experiments [17]. The powder of gardenia was extracted using 60% ethanol at a solid–liquid ratio of 1:12 (g mL^−1^) at 60 °C for 300 min. The supernatant was then concentrated using a rotary evaporator to obtain the viscous infusion. The 1:7 volume ratio of the petroleum ether was added to the viscous infusion, and the crude extract of gardenia (EGJ) was acquired after the concentration and vacuum freeze-drying processes.

According to the results of the preliminary experiments, the elution effect of the AB-8 macroporous adsorption resin was the best [19]. The crude EGJ was loaded onto AB-8 resin columns (25 mm × 500 mm) and eluted with 2 BV of water (1 BV = 1500 mL) and 20%, 40%, and 60% ethanol successively. Next, the samples were obtained by rinsing them with a 20% ethanol solution, 40% ethanol solution, and 60% ethanol solution, respectively. Subsequently, the 20% EGJ and 40% EGJ eluents were separately collected at 500 mL intervals for the relatively single components, which were numbered in the order of the sample from No.1 to No.30 and analyzed using the absorbance value measurement. According to the characteristic wavelength, the No.4, No.8, No.9, and No.14 samples were analyzed using HPLC. The Waters module 2695 was equipped with a 2998 photodiode array detector (Milford, MA, USA) and a Diamonsil Plus C18-A column (250 mm × 4.6 mm, 5 µm, Dikma, Shanghai, China). The column temperature was kept at 30 °C and the injection volume was 20 µL. An elution using solvent A (0.5% acetic acid) and solvent B (methanol) in a step gradient manner at a flow rate of 1 mL min^−1^ was carried out as follows: 0–10 min, 10–20% B; 10–20 min, 20–50% B; 20–30 min, 50–50% B; 30–40 min, 50–70% B; 40–50 min, 70–95% B; 50–60 min, 95–95% B; 60–65 min, 95–10% B; 65–75 min, 10–10% B. The detection wavelength was set to 238 and 440 nm. 

In addition, the structures of the geniposide and crocetin were identified following an optimized method [7,20,21]. To obtain the structure of the geniposide and crocetin in detail, the UPLC-ESI-QTOF-MS/MS system was performed on an ACQUITY UPLC instrument connected to a Xevo G2Q-TOF mass spectrometer via an ESI interface (Waters Corp, Milford, MA, USA). For the qualitative analysis, an analytical column (100 mm × 2.1 mm, 1.7 μm) was used to separate the complex fraction compounds. The injection volume was 5 μL, consisting of a 0.1% formic acid solution (A) and acetonitrile (B) as the mobile phase. The flow rate was 0.4 mL/min and the gradient elution conditions were 0 min, 1% B; 1 min, 1% B; 25 min, 99% B; 27 min, 99% B; and 30 min, 1% B. In the meantime, the column temperature was 30 °C, the voltage was 550 V, the dry gas temperature was 500 °C, the atomizing gas was N_2_, the pressure was 50 psi, and the mass-to-charge ratio was 50–1500 *m*/*z*.

### 2.3. Inhibition of Enzyme Activity In Vitro

Based on the characteristic wavelengths of the samples, the No. 4, No. 8, No. 9, and No. 14 samples were used for the in vitro α-glucosidase activity inhibition assays. The α-glucosidase inhibition assays were carried out following the method established by Mojica et al. (2015) [22]. In brief, 40 μL of the samples were added to 20 μL of a 0.25 unit/mL yeast α-glucosidase solution. The α-glucosidase was incubated in 0.1 M of a sodium phosphate buffer (PBS) (pH 6.9) for 15 min at 37 °C in a 96-well plate. Then, 40 μL of the 0.5 mM *p*NPG in a PBS was added to each well and incubated at 37 °C for 20 min. The reaction was stopped using 100 μL of 0.1 M Na_2_CO_3_. Acarbose was used as the positive control and the PBS was used as the blank control. In the reaction system, a half-maximal inhibitory concentration (IC_50_) of the No.8 and No.9 samples was performed by fixing the concentration of the enzyme (0.25 U mL^−1^) and substrate *p*NPG (0.5 mol L^−1^). The inhibitory activity was expressed as the percentage inhibition of the enzyme activity and was calculated using Equation (1).
(1)$$Inhibitory\;activity\;(\%)=\frac{A_{sample}-A_{k2}}{A_{control}-A_{k1}}\times 100\%$$
where *A_control_* is the absorbance of the PBS that replaces the samples, *A_sample_* is the absorbance of the samples, *A_k_*_1_ is the absorbance of the PBS that replaces the samples and the enzyme, and *A_k_*_2_ is the absorbance of the PBS that replaces the enzyme [23].

### 2.4. Animal Experiment

The 8-week-old SPF male Kunming mice were purchased from the Experimental Animal Center of Huazhong Agricultural University (Certificate Number: SCXK (Hubei) 2015-0019, Wuhan, China). The animal experiment was carried out at the Experimental Animal Center of Huazhong Agricultural University with the approval of the Scientific Ethics Committee (Permission No. HZAUMO-2018-068). All the experimental operations were performed in strict accordance with the experimental animal standards established by the People’s Republic of China (GB 14925-2010) and the guidelines for animal feeding and use during the experiment of the Experimental Animal Center of Huazhong Agricultural University (license number: No. SYXK (Hubei) 2015-0084). This study was conducted according to the standards in the Guiding Principles in the Care and Use of Animals. During the experimental period, the mice were housed in each cage under external environmental conditions of a temperature of 24 ± 1 °C, a humidity of 40 ± 10%, and a normal light/dark cycle (12 h/12 h). Throughout the experiment, the animals had free access to distilled water and standard laboratory pellets. They were acclimatized and fed for one week before the start of the experiment. 

#### 2.4.1. Establishment of the Type 2 Diabetes Mellitus (T2DM) Mouse Model and Drug Administration

According to the adaptive feeding situation and the weight of the mice, nine mice were chosen as the normal control group and continued to be fed using an ordinary diet. The other mice were fed using a high-fat diet [24]. Each group consisted of nine mice. Four weeks later, the mice in the high-fat diet group were intraperitoneally injected with streptozotocin (STZ, 65 mg kg^−1^bw^−1^) for three consecutive days [6]. Then, the mice continued to be fed using a high-fat diet for one week. The fasting blood glucose levels of the mice were measured after fasting for 12 h. The group with a fasting blood glucose value ≥ 7.8 mmol/L could be regarded as the successful T2DM mice. They were divided into a normal control group (NC group), a T2DM group (DM group), a metformin positive control group (M group, 200 mg kg^−1^), a 20% EGJ group (100 mg kg^−1^), a 40% EGJ group (200 mg kg^−1^), and an EGJ group (600 mg kg^−1^). The mice in each group were fed for 5 weeks. The flow diagram of the animal experiments is shown in Figure 1.

#### 2.4.2. Determination of the Body Weight, Food Intake, Water Intake, Blood Glucose, and Oral Glucose Tolerance

The mice were weighed daily, and their food and water intake were recorded. The fasting blood glucose (FBG) was measured once a week until the blood glucose level of the experimental group showed a downward trend. The oral glucose tolerance test (OGTT) was performed 2 days before the end of the animal experiment. The test procedure was as follows. After fasting for 12 h, the first FBG value was determined, recorded as BG_0_, and then the glucose solution was perfused at a dose of 1 g kg^−1^bw^−1^. The test time for each mouse was strictly controlled, and the glucose was measured at 15 min, 30 min, 60 min, and 120 min after the gavage, respectively. The values were recorded as BG_15_, BG_30_, BG_60_, and BG_120_. The blood glucose curve was drawn for each group and then the area under the curve (AUC) for the glucose was calculated.

#### 2.4.3. Biochemical Assays and the Detection of the Organ Indexes

The retro-orbital site was used for the blood sampling. The mice were euthanized using cervical dislocation and were quickly dissected to obtain the livers and the kidneys. Within 24 h, the blood was centrifuged at 3000 r/min for 15 min. The serum was sub-packaged and stored at −20 °C for the detection of the glycosylated serum protein (GSP), total cholesterol (TC), triglyceride (TG), high-density lipoprotein cholesterol (HDL-C), and low-density lipoprotein cholesterol (LDL-C). The atherosclerosis index (AI) and cardiovascular risk index (CVRI) of the lipid metabolism were calculated using Equations (2) and (3) [24]. SOD and CAT belong to the antioxidant enzymes, which play a key role in the oxidation and antioxidant balance of the body and can protect the integrity of the β cell structure and function. MDA is the final peroxidation product of the lipids, and its expression level can reflect the level of oxygen free radicals in the body and measure the damage degree of the oxidative stress, which has been proven to be an important cause of inducing apoptosis of the pancreatic β cells and inhibiting the insulin secretion. For the assays of the total protein, hepatic malondialdehyde (MDA), catalase (CAT), and superoxide dismutase (SOD), the liver tissues were homogenized to a 10% liver homogenate and centrifuged at 4000 r/min for 10 min at 4 °C. Then, the supernatant was obtained. All the biochemical indexes were determined using the detection kits. The liver and kidney indexes were calculated according to Equation (4).
(2)$$AI=\frac{LDL-C}{HDL-C}$$
(3)$$CVRI=\frac{TG}{HDL-C}$$
(4)$$Organ\;index\;(\%)=\frac{Orgain\;weight\;(\mathrm{mg})}{Weight\;of\;mice\;(g)}\times 100\%$$

### 2.5. Statistical Analysis

The data were expressed as the mean ± the standard deviation (SD). The SPSS Statistics 22.0 was used to analyze the significant difference between the logarithm data of the one-way analysis of variance (ANOVA) and Duncan’s multiple range (DMRT). When the value of *p* < 0.05, there was a statistical difference between the judgment data. The GraphPad Prism 5 software was used to draw the corresponding charts for the data analysis results.

## 3. Results and Discussion

### 3.1. HPLC Analysis of Geniposide and Crocetin

As shown in Figure 1 and Figure 2, the HPLC chromatograms of the crude EGJ and its fractions were recorded at λ = 238 nm or 440 nm. The samples were prepared according to the above method. The peak of the geniposide standard was recorded at 238 nm and its retention time was 20.72 min. The characteristic wavelength of the No.4 sample was 238 nm and the retention time was as same as the geniposide standard. Therefore, the No.4 sample mainly included geniposide. By the same argument, the retention time of crocin Ⅰ was 28.67 min with a corresponding wavelength of 440 nm. Thus, the No.8 and No.9 samples mainly contained crocin I. The peak of the 20% EGJ fraction was recorded at 238 nm, which implied that the 20% EGJ fraction mainly contained geniposide. In addition, the crocin I standard and geniposide standard were taken as the references for the quantitative analysis. The two linear regression equations for geniposide and crocin I were y = 1.0664x + 1.4066 (R^2^ = 0.9994) and y = 1.1889x − 10.92 (R^2^ = 0.9959). The peak areas of geniposide and crocin I from gardenia were 57.603 ± 0.93 and 58.481 ± 1.25. According to the above formula, the contents of geniposide and crocin I in gardenia were 53.541 ± 1.17 mg g^−1^ and 57.679 ± 0.95 mg g^−1^. This result laid the groundwork for the further screening of the four fractions on the α-glucosidase activity in vitro.

### 3.2. Identification of Geniposide and Crocetin Using UPLC-ESI-QTOF-MS/MS

Figure 3A–C show the pattern of total ion chromatogram (TIC) of the UPLC-ESI-MS for the geniposide and crocin I standards, the EGJ, and the No.9 sample. As shown in Figure 3A, the peak times for geniposide and crocin I was recorded at 6.76 min and 8.60 min, respectively. As shown in Figure 3B, the two peaks of crocin I were recorded at 8.54 min and 10.81 min. Since the peak of the crocin I standard was recorded at 8.60 min, the former should be crocin I, while the latter might be the isomer C_44_H_64_O_24_. The crude EGJ was captured with 12 peaks, including geniposide and crocin I. As shown in Figure 3C, the No. 9 sample was recorded with 18 peaks, including geniposide and crocin I. Meanwhile, the size of the peak indicated that it contained a large amount of crocin I and a very small amount of geniposide. The mass cracking pathways of these two main compounds could be inferred based on the MS and MS–MS spectrum information.

According to the primary and secondary mass spectra of geniposide, the mass-to-charge ratio (*m*/*z*) of geniposide with sodium ([M+Na]^+^) was 411.1267, the two geniposide couples ([2M+H]^+^) corresponded to 777.2850, the geniposide ([M+H]^+^) was 389.1268; the geniposide stripped of one molecule of monosaccharide ([M-glc+H]^+^) was the fragment ion 227.0921, and the geniposide stripped of one molecule of monosaccharide and one molecule of water ([M-glc-H_2_O+H]^+^) was 209.0805, as shown in Figure 4A. It was presumed that it was relatively easy for geniposide to remove one molecule of monosaccharide, followed by a further removal of one molecule of water from the parent nucleus of the cyclic enol ether terpene. Its mass spectrometry cleavage pathway is shown in Figure 4A.

On the basis of the primary and secondary mass spectra of crocin I, the mass-to-charge ratio (*m*/*z*) of crocin I ([M+Na]^+^) with sodium ions was 999.3741, crocin I ([M+H]^+^) corresponded to 976.3801; crocin I with the sodium ions stripped of the disaccharides ([M-2glc+Na]^+^) was the fragment ion 675.2639, crocin I stripped of two disaccharides ([M-4glc]^+^) was 329.1755, and one molecule of disaccharide with the sodium ions ([M-2glc-C_20_H_20_O_3_]^+^) was 347.0957, as shown in Figure 4B. It was presumed that it was relatively easy to break the glycosidic bond attached to the parent chain from the disaccharide at both ends of crocin I. Its mass spectral cleavage pathway is shown in Figure 4B.

Based on the time of each peak and their mass spectrometry information, sixteen compounds were identified, consisting of flavonoids, iridoid, organic acids, and diterpene, as shown in Table 1.

### 3.3. Screening of Geniposide and Crocetin on the α-Glucosidase Activity In Vitro

As shown in Figure 5A, the different compounds had different inhibitory effects on the α-glucosidase at the same concentration. The concentration of the No. 8 and No. 9 samples was 0.5 mg mL^−1^ and inhibition rates were 35.74 ± 4.86% and 97.61 ± 8.12%, respectively. At the same concentration as the No. 8 and No. 9 samples, the No.14 sample (21.79 ± 4.27%) had a slightly weaker inhibition ability than the acarbose (35.99 ± 3.28%). Therefore, both the No. 8 and No. 9 samples had a better ability of inhibiting the enzyme. The IC_50_ of the No. 8 and No. 9 samples were tested to further screen their inhibition ability in vitro. As shown in Figure 5B, the IC_50_ of the No.8 and No.9 samples were 0.577 ± 0.041 mg mL^−1^ and 0.204 ± 0.007 mg mL^−1^, respectively. This result indicated that the enzyme inhibitory effect of the No.9 sample was the best. Since the No. 4 sample was obtained through the elution of a 20% ethanol concentration and the No. 9 sample was obtained through the elution of the 40% ethanol concentration, the No.4 and No.9 samples as well as the crude EGJ were named as the 20% EGJ, 40% EGJ, and EGJ group, respectively, and used for the subsequent animal experiments.

#### 3.3.1. Type of Inhibition for the No.9 Sample

As mentioned before, the No. 9 sample showed the best inhibition of the enzyme, so this sample was used to study the type of inhibition. Using the enzyme concentration as the horizontal coordinate and the reaction rate as the vertical coordinate, the straight lines were obtained for the different concentrations of the No. 9 sample (0.18–0.22 mg/mL). The results are shown in Figure 6, where each straight line passed through the origin and the slope was negatively correlated with the concentration of the sample, i.e., the type of inhibition of the α-glucosidase for the No. 9 sample was reversible and the enzyme inhibited by the sample could be revived using physical methods, such as dialysis and ultrafiltration [25].

#### 3.3.2. Mechanism of Inhibition for the No.9 Sample

As stated before, the type of inhibition of the No.9 sample was reversible, and this sample was then subjected to a kinetic study of the inhibition. Using the reciprocal of the pNPG concentration as the horizontal coordinate and the reciprocal of the reaction rate as the vertical coordinate, the Lineweaver–Burk plot was used to obtain a straight line for the different concentrations of the No.9 sample (0.06–0.14 mg/mL), which was judged by the size of the intercept and the position of the coordinate. The results are shown in Figure 7. When the sample concentration increased, the vertical coordinate of the corresponding line grew, indicating that the 1/V increased and the V_max_ decreased. A decrease in the horizontal coordinate indicated a decrease in −1/K_m_ and an increase in the Mie’s constant K_m_. According to the principle of the enzymatic reaction, it was clear that the No. 9 sample was anti-competitive for the α-glucosidase inhibition [26].

### 3.4. Effects of the Crude EGJ, 20% EGJ, and 40% EGJ in the DM Male Mice

#### 3.4.1. Effects of the Crude EGJ, 20% EGJ, and 40% EGJ on the Fasting Blood Glucose, Body Weight, and Water Intake

The weight changes of the mice after three days of the STZ injection were shown in Figure 8. The weight of the mice in the general group increased slowly while the weight of the mice in the model group increased on the second day and then decreased continuously. Until the eighth day, the weight of the model group dropped to 37.3 ± 1.6 g, while the weight in the general group increased to 44.0 ± 1.1 g. At the same time, the level of the blood glucose in the different groups was measured. The level of the blood glucose in the general group and the model group fluctuated between 4.67 mmol/L and 19.14 mmol/L, respectively. In addition, it was found that the hair of the mice in the general group was normal white and the mice were active, while the hair of mice in the model group w yellow, wet, and the mice behaved lazily, indicating that the DM modeling was successful.

The weight changes of the mice in each group during the administration period is shown in Figure 9A. The weight of the mice in the NC group always maintained the highest level and constantly increased. The initial body weights of the mice in the other groups were lower and later changed to varying degrees. The body weight of the mice in the DM group decreased, while the mice in the M group did not change much, indicating that the weight of the T2DM mice could be controlled using metformin. Compared to the EGJ group and the 40% EGJ group, the weight in the 20% EGJ group maintained an upward trend and increased to approx. 42 g at the end of the fifth week, indicating that the 20% EGJ (geniposide) had a certain effect on controlling the weight loss caused by diabetes.

The quantity of the water and food intake of the mice in each group during the administration period are shown in Figure 9B,C. The water intake in the 20% EGJ group also decreased, and the quantity of the food intake was kept at a low level at the end of the experiment, at approx. 11.01 g/unit. The results suggested that metformin and geniposide had a good effect on adjusting the quantity of the water and food intakes of the T2DM mice, and that geniposide had the highest efficacy in this study.

#### 3.4.2. Effects of the Crude EGJ, 20% EGJ, and 40% EGJ on the FBG, OGTT, and AUC

As shown in Figure 10A, during the period of administration, the level of the FBG in the NC group was basically stable at the normal level (≤7.8 mmol/L). Compared to the NC group, the FBG in the other groups were significantly higher. The FBG in the DM group was the highest, indicating that the hyperglycemia symptoms were the most severe [27]. In the M group, the level of the FBG reached the highest level in the first week, and then decreased, indicating that the metformin could restore the FBG of the T2DM to fluctuate within the normal range. There was no significant difference between the 20% EGJ group and the EGJ, for which the FBG levels were separately 11.06 ± 2.62 and 17.33 ± 2.25 mmol/L at the fifth week. However, the 40% EGJ group did not show a significant downward trend. These results indicated that the metformin, 20% EGJ (geniposide), and EGJ could effectively alleviate the level of the FBG in the T2DM mice.

The OGTT experiment was carried out two days before the end of the experiment. As shown in Figure 10B, the level of the blood glucose of the mice in each group increased after 15 min caused by the glucose intake on fasting. Except for the DM group, the level of the blood glucose in the other groups recovered at 2 h in varying degrees, and better recoveries were provided in the metformin and 20% EGJ groups. According to the AUC curve (Figure 10C), geniposide and metformin had the same effect for impairing the glucose tolerance, which was 63.52% lower than the DM group. The value of the AUC in the 40% EGJ and EGJ groups decreased by 39.17% and 50.06%, respectively. However, the results of the postprandial blood glucose were unstable, and more results were needed.

#### 3.4.3. Effects of the Crude EGJ, 20% EGJ, and 40% EGJ on the Organ Indexes and GSP

The organ indexes in the NC group were the lowest, and the indexes in the DM group were the largest, indicating that diabetes caused hepatomegaly and liver injury in the mice. Metformin could alleviate the liver injury better than the 20% EGJ, and there was no significant difference between metformin and the 20% EGJ (Table 2). The GSP reflected the blood glucose concentration in the first two weeks. The GSP index of the mice in each group after 5 weeks of administration is shown in Figure 11. The level of the GSP in the NC, M, and 20% EGJ groups were significantly different compared to the DM and 40% EGJ groups (*p* < 0.05). The results showed that the 40% EGJ (crocin I) and 20% EGJ (geniposide) groups both had a certain effect for reducing the GSP index in vivo, while the geniposide fraction was better.

#### 3.4.4. Effects of the Crude EGJ, 20% EGJ, and 40% EGJ on the Serum TC, TG, HDL-C, LDL-C, AI, and CVRI 

After the drug administration, the level of the TC of the mice in the NC group, the DM group and the sample group altered remarkably with significant differences (*p* < 0.05), as shown in Figure 12A. The TC levels of the sample groups all decreased by approx. 32% compared to the DM group. As shown in Figure 12B, the decrease in the levels of the TC was similar in the M and the 20% EGJ groups, which was 15% lower than the NC group. Compared to the DM group, the EGJ group showed a significant decrease (*p* < 0.05). However, the 40% EGJ group failed to increase the level of the TC. As shown in Figure 12C, the HDL-C levels were significantly increased in all the sample groups, except for the 40% EGJ group. The HDL-C levels in both the M and the 20% EGJ groups were approximately 56% higher than those in the DM group. The abnormal lipid metabolism symptoms of the sample group mice were improved since the HDL-C contributed to the removal of the cholesterol. As shown in Figure 12D, the level of the LDL-C in the M group returned to normal, and the decrease in the LDL-C in the 20% EGJ group was the most obvious; the value was 17.55% lower than NC group. Since an excessive LDL-C can lead to cholesterol accumulation, the administration group could improve the abnormal lipid metabolism of the mice. In addition, the level of the LDL-C in the 40% EGJ group was similar to the M group. As shown in Figure 12E,F, the AI and CVRI indexes were the highest in the DM group, indicating the highest risk index for atherosclerosis and cardiovascular disease. The index was significantly lower in the M group, 20% EGJ group, 40% EGJ and crude EGJ group. The above results showed that the 20% EGJ (geniposide) group showed an obvious improvement effect in the T2DM mice.

#### 3.4.5. Effects of the Crude EGJ and Its 20% and 40% Fractions on Hepatic MDA, CAT, and SOD

As shown in Figure 13A, the MDA levels in the M and 20% EGJ groups were reduced. However, there was no significant difference between the MDA levels in the 40% EGJ group and the model group, indicating that the 40% EGJ (crocin I) group did not show an ability to eliminate the free radicals in the DM mice, whereas the 20% EGJ (geniposide) group could scavenge the free radicals in the DM mice. Then, we further measured the level of CAT. As shown in Figure 13B, there was no significant difference between the 40% EGJ group and the DM group. The level of CAT in the other groups could not reach the level in the NC group but was higher than the DM group, indicating that metformin and the 20% EGJ could improve the level of CAT in the T2DM mice. Consequently, metformin and the 20% EGJ (geniposide) could accelerate the decomposition of hydrogen oxide to produce oxygen and water in the body and reduce the damage to the body. SOD could increase the active substances in the body and reduce the harmful substances produced by the metabolism. The level of SOD was similar to the level of MDA and CAT. The 40% EGJ group failed to improve the level of SOD the mice, but the level of SOD in the other groups of mice were increased [28], and the effect of the 20% EGJ (geniposide) was better than the M group. The above showed that geniposide could improve the oxidative stress damage in the T2DM mice and reduce its damage to the pancreatic islet β cells. The geniposide has the best effect and the extent of the compounds on T2DM followed the order of geniposide > crocin I > crude EGJ.

## 4. Conclusions

The hypoglycemic mechanism of the α-glucosidase inhibitors showed that, by inhibiting α-glucosidase in the intestinal mucosa, the breakdown of starches into glucose was slowed, reducing and delaying the absorption of glucose in the small intestine to lower the blood glucose, which was more obvious for postprandial hyperglycemia [29,30]. The main α-glucosidase inhibitors commonly used were acarbose and voglibose [31,32]. Related studies showed that gardenia can effectively inhibit the activity of the α-glucosidase [33]. However, there has been little comprehensive comparison between the hypoglycemic effects of geniposide, crocin I and crocin II or in vitro and in vivo evaluation. In this chapter, a type 2 diabetes mellitus (T2DM) mice model was induced using a high-fat diet plus an intraperitoneal injection of streptozocin (STZ). A crude extract of gardenia (EGJ), geniposide, and crocin I was administrated to the mice. The high-fat diet plus the intraperitoneal injection of STZ induced the rapid decreases in the body weights with a yellow and wet hair color. Approx. 10 days after the establishment of the diabetes model, the body weight of the mice in the model group was as low as 37.3 ± 1.6 g, the water intake and food intake were as high as 44.73 mL/units and 22.47 g/units, respectively, and the blood glucose level fluctuated around 19.14 mm/L. The mice in the model group showed the lowest body weight and the highest water and food intakes and blood sugar levels compared to the mice in the normal group, proving that the diabetes model was successful. The mice in the administration metformin positive drug and administration 20% EGJ groups were accompanied by a higher body weight gain, a lower water and food intake, and a lower blood glucose level the T2DM group. The protective effect of the 20% EGJ was similar to metformin in the diabetic mice from the liver injury. Compared to the model group, the 20% EGJ, 40% EGJ, and crude EGJ all could decrease the serum GSP, TC, TG, HDL-C, and other indicator concentrations, among which the 20% EGJ had the most significant effect for regulating the glucolipid metabolism in the T2DM mice. The main content of the 20% EGJ was geniposide, indicating that geniposide had a certain effect on the glucose metabolism and the lipid metabolism in the diabetic mice. The effect of geniposide for treating T2DM was much higher than that of crocin I in vivo. However, the main component of the No. 8 sample (IC_50_ = 0.577 ± 0.041 mg ml^−1^) was geniposide, which was a weak inhibitor of the α-glucosidase in vitro. The component of the No. 9 sample (IC_50_ = 0.204 ± 0.007 mg ml^−1^) was crocin I, which had a stronger α-glucosidase inhibitory activity in vitro compared to the No.8 sample. This different result was not contradictory to the in vivo results. The studies suggested that the hypoglycemic effect of the alleviated geniposide might have been mediated by alleviating the glycogen phosphorylase and glucose-6-phosphatase activities [34]. It was also found that geniposide could significantly upregulate the protein levels of Glc NAc T-Ⅳ, a glycosyltransferase, but had no significant effect on the expression of clathrin [35]. Some studies found that gardeniside could stimulate the glycogen synthesis in the mice with a high-fat diet and a streptozotocin injection [36]. Therefore, in this research, the result of geniposide was effective in vivo but not in vitro, which also complemented the mechanism that the primary target of the geniposide participating in the treatment of T2DM was not α-glucosidase. Additionally, the action target of crocin I was α-glucosidase in vitro. The evidence reported that crocin I had benefits on the glycemic parameters and the type 2 diabetes-related complications in vivo [37,38]. However, there has been minimal research on its targets. This paper added that the target for the treatment of crocin I was α-glucosidase. At the same time, this experiment proved that geniposide was better than crocin I for the regulation of the blood glucose and blood lipids in the diabetic mice.

In conclusion, the separated and purified monomers of the geniposide and crocetin compounds from the crude extract of gardenia (EGJ) were collected and analyzed using HPLC. The monomers of the geniposide and crocin I compounds were compared using their hypoglycemic effects in vitro and in vivo. The extent of the compounds on T2DM followed the order of geniposide > crocin I > crude EGJ. The findings identified the different mechanism between crocin I and geniposide. Crocin I was effective on T2DM by inhibiting the α-glucosidase, geniposide played a major role on the hypoglycemic effect in the apparent levels of glucose, the lipid metabolism, and the apparent level of oxidative stress in the mice. This paper laid a further experimental foundation for the development of the crude EGJ as a new nutritional supplement and provided a theoretical reference for the high value utilization of the crude EGJ. Further studies are needed to compare the hypoglycemic mechanisms between the geniposide, crocin I, and EGJ.

## Data Availability

Data is contained within the article.

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
