# Peer review of "Effective Therapeutic Verification of Crocin I, Geniposide, and Gardenia (Gardenia jasminoides Ellis) on Type 2 Diabetes Mellitus In Vivo and In Vitro"

_foods, 2023, doi:10.3390/foods12081668_

Round 1

Author Response

Thank you all so much for your reviewing my work (Manuscript ID:foods-2297227, Title: Therapeutic effective verification of crocin I, geniposide and gardenia (Gardenia jasminoides Ellis) on type 2 diabetes mellitus in vivo-in
vitro).

First, I want to show my respect to all the reviewers, because they are competent, and their comments and suggestions are highly reasonable, and help me a lot.

According to their comments, I have tried my best to revise my manuscript. And now, I will give my detailed explanations for the revision:

Editorial comments:

Reviewer 1

(1)The manuscript needs severe editing, rewording and remediation of multiple grammatical errors, that, unfortunately, in many cases, make the text incomprehensible. Some of them can be found in the fragments below, that must be rewritten:

72-73. The objective of present study was to compare and identify the structure of bioactive compounds of gardenia in vivo and vitro experiments.

Response:

The objective of the present study was to compare the hypoglycemic effects of bioactive compounds from gardenia in in vivo and in vitro experiments and to identify the structural of them. According to your suggestion, the description has been added on Page 6, Line 101-103.

73-75. Crude extract of gardenia (Gardeniajasminoides Ellis) (EGJ) was isolated and purified to obtain geniposide and crocin I by different concentration of alcohol.

Response:

Different concentrations of alcohol were used to isolate and purify the crude extracts of gardenia (Gardenia jasminoides Ellis) (EGJ), and finally geniposide as well as crocin I were obtained. According to your suggestion, the description has been added on Page 6, Line 105-107.

75-76. It was explored that the hypoglycemic effect of geniposide, crocin I and EGJ intervention role playing on T2DM.

Response:

This paper explored that the hypoglycemic effect of geniposide, crocin I and EGJ intervention role playing on T2DM. According to your suggestion, the description has been added on Page 6, Line 107-109.

99-102. Crude extract of gardenia (EGJ) was obtained by adding petroleum ether in a volume ratio of 1:7 after concentrating and vacuum freeze-drying. Approximately 0.8 g of freeze-dried crude EGJ was dissolved in 100 mL water for isolating.

Response:

The powder of gardenia was extracted with 60% ethanol at a solid-liquid ratio of 1:12 (g mL-1) at 60 °C for 300 minutes and the supernatant was then concentrated by rotary evaporator to obtain viscous infusion. Add 1:7 volume ratio of petroleum ether to the viscous infusion, and acquire the crude extract of gardenia (EGJ) after concentrating and vacuum freeze-drying. According to your suggestion, the description has been added on Page 7, Line 130-134.

120-121. Samples of No.4, No.8, No.9 and No.14 were used to do experiment of α-glucosidase active inhibition in vitro.

Response:

Thank you very much for the constructive suggestions. The fragment does not belong to the section 2.2. We deleted this sentence in the revised manuscript.

126-127. Besides, structures of geniposide and crocetin were identified with the modified method [7, 18, 19]

Response:

In addition, the structures of geniposide and crocetin were identified following an optimized method. According to your suggestion, the description has been added on Page 8, Line 152-153.

139-140. According to the characteristic wavelength of samples, No.4, No.8, No.9, No.14 were performed to do experiment of α-glucosidase active inhibition in vitro.

Response:

Based on the characteristic wavelengths of the samples, samples No. 4, No. 8, No. 9 and No. 14 were used for in vitro α-glucosidase activity inhibition assays. According to your suggestion, the description has been added on Page 9, Line 165-166.

182-183. All groups of mice were administered continuously by gavage for 5 weeks.

Response:

The mice in each group were administrated intragastric for 5 weeks. According to your suggestion, the description has been added on Page 11, Line 210.

  1. Recording the daily intake of food and water.

Response:

The mice were weighed daily and their food and water intake were recorded. According to your suggestion, the description has been added on Page 12, Line 217.

192 Oral glucose tolerance test (OGTT) was measured before two days of the end.

Response:

The oral glucose tolerance test (OGTT) was performed 2 days before the end of the animal experiment. According to your suggestion, the description has been added on Page 12, Line 219-220.

197-198. Drawing the blood glucose curve of each group. Then, calculating the area under the curve (AUC) of glucose.

Response:

The blood glucose curve was drawn for each group and then the area under the curve (AUC) for glucose was calculated. According to your suggestion, the description has been added on Page 12, Line 225-226.

226-235. As shown in Fig. 1A and Fig. 1B, the HPLC chromatograms of crude EGJ and its fractions should be recorded at λ = 238 nm or 440 nm. Samples were prepared according to above methods, and single fragments of the compound were obtained. The peak of geniposide standard recorded at 238 nm and the retention time of it was 20.72 min. The characteristic wavelength of No.4 sample was 238 nm and the retention time was as same as the geniposide standard products. Therefore, No.4 sample mainly included geniposide. By the same argument, retention time of the crocin Ⅰ was 28.67 min with corresponding emission of 440 nm. Thus, No.8 sample and No.9 sample mainly contained crocin I. The peak of 20% EGJ component recorded at 238 nm, which means the main compound in the 20% EGJ component was geniposide.

Response:

As shown in Fig. 1A and Fig. 1B, the HPLC chromatograms of crude EGJ and its fractions were recorded at λ = 238 nm or 440 nm. Samples were prepared according to the above method. The peak of geniposide standard was recorded at 238 nm and the retention time of it was 20.72 min. The characteristic wavelength of No.4 sample was 238 nm and the retention time was as same as the geniposide standard. Therefore, No.4 sample mainly included geniposide. By the same argument, retention time of the crocin Ⅰ was 28.67 min with corresponding wavelength of 440 nm. Thus, No.8 sample and No.9 sample mainly contained crocin I. The peak of the 20% EGJ fraction was recorded at 238 nm, which implies that the 20% EGJ fraction contains mainly geniposide. According to your suggestion, the description has been added on Page 13-14, Line 254-262.

256-260. As shown in Fig. 2A, different fragments had different inhibitory effects on α-glucosidase at the same concentration. No.8 and No.9 sample were at a concentration of 0.5 mg mL-1 and inhibition rates were 35.74 ± 4.86% and 97.61 ± 8.12%, respectively. No.14 sample (21.79 ± 4.27%) had a slightly weaker inhibition ability than acarbose (35.99 ± 3.28%). Therefore, the No.8 and No.9 samples both had better inhibitory ability to enzyme.

Response:

As shown in Fig. 4A, different compounds had different inhibitory effects on α-glucosidase at the same concentration. The concentration of samples No. 8 and No. 9 was 0.5 mg mL-1 and inhibition rates were 35.74 ± 4.86% and 97.61 ± 8.12%, respectively. No.14 sample (21.79 ± 4.27%) had a slightly weaker inhibition ability than acarbose (35.99 ± 3.28%). Therefore, both samples No. 8 and No. 9 had a better ability of inhibiting the enzyme. According to your suggestion, the description has been added on Page 26, Line 382-390.

260-261. IC50 of them was tested with to further screen their inhibitory ability in vitro.

Response:

The IC50 of samples No. 8 and No. 9 were tested to further screen the inhibition ability of them in vitro. According to your suggestion, the description has been added on Page 26, Line 390-391.

  1. “the weight in model group loss to 37.3 ± 1.6 g”.

Response:

the weight of the model group dropped to 37.3±1.6 g. According to your suggestion, the description has been added on Page 30, Line 448.

319-320. “the mice was strong in sport”, “the mice was slow in sport”.

Response:

“the mice was active in sport”, “the mice behaved lazily” According to your suggestion, the description has been added on Page 30, Line 452-454.

324-326. “The initial weight of the mice in the other groups is at the low level that were different degrees of changes later. The weight of mice was lost in the DM group, while there was little change in the M group”

Response:

The initial body weight of mice in the other groups was lower and later changed to varying degrees. The body weight of mice in the DM group decreased, while that of mice in the M group did not change much. According to your suggestion, the description has been added on Page 30, Line 457-459.

  1. “indicating the disease was the most serious”

Response:

indicating the hyperglycemia symptoms were most severe. According to your suggestion, the description has been added on Page 33, Line 486.

359-360. “40% EGJ group was no obvious downward trend”

Response:

the 40% EGJ group did not show a significant downward trend. According to your suggestion, the description has been added on Page 33, Line 491-492.

379-380, 392-393 425-426, 453-454. “and different letters marked above the bars were significantly different”

Response:

and different letters marked above the bars showed significant differences. According to your suggestion, the description has been added on Page 34, Line 511-512;Page 35, Line 525;Page 38, Line 558-559;Page 40, Line 583-584.

  1. “After pharmacodynamics intervention”

Response:

After drug administration. According to your suggestion, the description has been added on Page 37, Line 536.

  1. “sample group was significantly difference”

Response:

sample group altered remarkably with significant differences. According to your suggestion, the description has been added on Page 37, Line 537.

404-405. “The sample groups all decreased about 32% lower than the DM group.”

Response:

The TC levels of sample groups all decreased by about 32% compared to the DM group. According to your suggestion, the description has been added on Page 37, Line 538-539.

408-409. “As played in Fig. 9C, the level of HDL-C could be significantly increased except 40% EGJ”

Response:

As shown in Fig. 11C, HDL-C levels were significantly increased in all sample groups except for the 40% EGJ group. According to your suggestion, the description has been added on Page 37, Line 542-543.

409-412.  “Both the M group and the 20% EGJ group were 56% higher than the DM group. According to HDL-C can promote the clearance of cholesterol, the abnormal lipid metabolism of mice in administration group could be improved.” It is not clear enough. What is higher? What promotes the clearance?

Response:

HDL-C levels in both the M and 20% EGJ groups were approximately 56% higher than those in the DM group. The abnormal lipid metabolism symptoms of the sample group mice can be improved because HDL-C contributes to the removal of cholesterol. According to your suggestion, the description has been added on Page 37, Line 543-546.

  1. “While the index could be significantly reduced”

Response:

While the index was significantly lower. According to your suggestion, the description has been added on Page 37, Line 553-554.

430-446. Check the text and rephrase

Response:

As shown in Fig. 12A, the MDA levels in the M and 20% EGJ groups were reduced, however, there was no significant difference between the MDA levels in the 40% EGJ group and the model group, indicating that 40% EGJ (crocin I) had no ability to eliminate free radicals in DM mice, but 20% EGJ (geniposide) could scavenge free radicals in DM mice. Then we further measured the level of CAT shown in Fig. 10B, there was no significant difference between the 40% EGJ group and the DM group. The level of CAT in the other groups could not reach the level in the NC group but higher than the DM group, indicating that metformin and 20% EGJ could improve the level of CAT of T2DM mice. Consequently, metformin and 20% EGJ (geniposide) could accelerate the decomposition of hydrogen oxide to produce oxygen and water in the body and reduce the damage to the body. SOD could increase the active substances in the body and reduce the harmful substances produced by metabolism, the level of SOD is similar to the level of MDA and CAT, 40% EGJ failed to improve the level of SOD in mice, but the level of SOD in the other groups of mice were increased[26], and the effect of 20% EGJ (geniposide) was better than the M group. The above showed that geniposide could improve the oxidative stress damage of T2DM mice and reduce its damage to pancreatic islet β cells, and the geniposide has the best effect and the extent of compounds on T2DM followed the order of geniposide > crocin I > crude EGJ. According to your suggestion, the description has been added on Page 39, Line 563-580.

457-474. Fig. 11 A-C was the pattern of Total Ion Chromatogram (TIC) of UPLC-ESI-MS of geniposide and crocin I standards, EGJ and No.9 sample. As shown in Fig. 11 A, the peak time of geniposide and crocin I was recorded at 6.76 min and 8.60 min respectively. As shown in Fig. 11 B, it was found two peaks recorded at 8.54 min and 10.81 min with crocin I. Since the standard product of crocin I peaked at 8.60 min, the former should be crocin I, while the latter might be the isomer of C44H64O24. It was found about 12 peaks recorded by crude EGJ, including those of geniposide and crocin I. As showed in Fig. 11 C, the TIC of No.9 sample was found about 18 peaks, including those of geniposide and crocin I. Meanwhile, the area of the peak explained it contained a large amount of crocin I and a very small amount of geniposide. The mass cracking pathways of these two main compounds could be inferred based on the MS and MS-MS spectrum information. For the geniposide with better effect in the in vivo hypoglycemic experiment, it was found that it preferentially lost one molecule of monosaccharide and then one molecule of water (Fig. 12 A); using the same method to resolve crocin I with better effect in the in vitro enzyme inhibition experiment, it was found that it was easier to lose the monosaccharide or disaccharide at both ends (Fig. 12 B). Based on the time of each peak and their mass spectrometry information, sixteen compounds were identified, mainly including flavonoids, iridoid, organic acids and diterpene, as shown in Table 2.

Response:

Fig. 2 A-C show the pattern of Total Ion Chromatogram (TIC) of UPLC-ESI-MS of geniposide and crocin I standards, EGJ and No.9 sample. As shown in Fig. 2 A, the peak time of geniposide and crocin I was recorded at 6.76 min and 8.60 min respectively. As shown in Fig. 2 B, the two peaks of crocin I were recorded at 8.54 min and 10.81 min. Since the peak of crocin I standard was recorded at 8.60 min, the former should be crocin I, while the latter might be the isomer of C44H64O24. The crude EGJ was captured with 12 peaks, including those of geniposide and crocin I. As showed in Fig. 2 C, sample No. 9 was recorded with 18 peaks, including those of geniposide and crocin I. Meanwhile, the size of the peak indicated it contained a large amount of crocin I and a very small amount of geniposide. The mass cracking pathways of these two main compounds could be inferred based on the MS and MS-MS spectrum information.

According to the primary and secondary mass spectra of geniposide, it is known that the mass-to-charge ratio (m/z) of geniposide with sodium ([M+Na]+) is 411.1267, 777.2850 corresponds to two geniposide couples ([2M+H]+) , 389.1268 is geniposide ([M+H]+); fragment ion 227.0921 is geniposide stripped of one molecule of monosaccharide ([M-glc+H]+), 209.0805 for geniposide stripped of one molecule of monosaccharide and one molecule of water ([M-glc-H2O+H]+), as shown in Fig. 3 A. It is presumed that geniposide is relatively easy to remove one molecule of monosaccharide, followed by further removal of one molecule of water from the parent nucleus of cyclic enol ether terpene, and its mass spectrometry cleavage pathway is shown in Fig. 3 A:

On the basis of the primary and secondary mass spectra of crocin I, the mass-to-charge ratio (m/z) of crocin I ([M+Na]+) with sodium ions is 999.3741, 976.3801 corresponds to crocin I ([M+H]+); fragment ions 675.2639 are crocin I with sodium ions stripped of disaccharides ([M-2glc+Na]+), 329.1755 for crocin I stripped of two disaccharides ([M-4glc]+), and 347.0957 for one molecule of disaccharide with sodium ions ([M-2glc-C20H20O3]+), as shown in Fig. 3 B. It is presumed that the disaccharide at both ends of crocin I is relatively easy to break the glycosidic bond attached to the parent chain, and its mass spectral cleavage pathway is shown in Fig. 3 B:

Based on the time of each peak and their mass spectrometry information, sixteen compounds were identified, consisting flavonoids, iridoid, organic acids and diterpene, as shown in Table 1.

According to your suggestion, the description has been added on Page 17-18, Line 282-313.

(2) Beware at the signification of the terms hypoglycemic and antihyperglycemic. Check all the article and use the correct terms.

Response:

We check all the article and use the correct terms. According to your suggestion, the description has been added on Page 5, Line 75.

(3) There are too many explanations/discussions in Section 3. Results. Keep only the results here and move the rest in Section 4. Discussion

Response:

I have changed the title from “Results” to “Results and discussion” According to your suggestion, the description has been added on Page 13, Line 252.

(4) Section “3.4. Identification of geniposide and crocetin by UPLC-ESI-QTOF-MS/MS” should be moved before in vitro and in vivo experiments, even if you explain on the basis of the obtained results the efficacy of some samples. Anyhow, those explanations should be moved in section 4.

Response:

We have moved section “3.4. Identification of geniposide and crocetin by UPLC-ESI-QTOF-MS/MS” before in vitro and in vivo experiments. According to your suggestion, the description has been added on Page 17-25, Line 281-382.

(5) Other correction needed:

 72-76. “The objective of present study was to compare and identify the structure of bioactive compounds of gardenia in vivo and vitro experiments. Crude extract of gardenia (Gardenia jasminoides Ellis) (EGJ) was isolated and purified to obtain geniposide and crocin I by different concentration of alcohol. It was explored that the hypoglycemic effect of geniposide, crocin I and EGJ intervention role playing on T2DM.” First, you identify the major bioactive components of the extract fractions using analytical methods, then, you compare the activity against enzyme and on mice. Extract of Gardenia was obtained and purified.

Response:

The objective of the present study was to compare the hypoglycemic effects of bioactive compounds from gardenia in in vivo and in vitro experiments and to identify the structural of them. First, the major bioactive components of the extract fractions were identified by using analytical methods. Then the activity of the major bioactive components against enzyme and on mice was compared. And different concentrations of alcohol were used to isolate and purify the crude extracts of gardenia (Gardenia jasminoides Ellis) (EGJ), finally geniposide as well as crocin I were obtained. According to your suggestion, the description has been added on Page 6, Line 101-107.

  1. “existing literature and preliminary laboratory experiments [17]”. If reference 17 indicates your laboratory experiments, other references should be included for “existing literature”.

Response:

The purified and modified fractions of gardenia were mainly prepared according to preliminary laboratory experiments[17]. According to your suggestion, the description has been added on Page 7, Line 129-130.

120-125. “Samples of No.4, No.8, No.9 and No.14 were used to do experiment of α-glucosidase active inhibition in vitro. Sample No.9 showed the best inhibition of α-glucosidase. Since sample No. 4 was obtained by elution of 20% ethanol concentration and sample No. 9 was obtained by elution of 40% ethanol concentration, samples No.4、 No.9 and crude EGJ were named as 20% EGJ、40% EGJ and EGJ group, respectively, and used for subsequent animal experiments.” This fragment does not belong to the section 2.2. Isolation and identification of geniposide and crocetin.

Response:

Thank you very much for the constructive suggestions. We deleted this fragment in the revised manuscript and this fragment was moved in section 3.2.

134-135. “An ESI electrospray ion source was used to capture molecular fragments in positive ion mode.” The ion source does not capture the fragments

Response:

To obtain the structure of geniposide and crocetin in detail, the UPLC-ESI-QTOF-MS/MS system was performed on an ACQUITY UPLC instrument connected with a Xevo G2Q-TOF mass spectrometer via ESI interface (Waters Corp, Milford, MA, USA). According to your suggestion, the description has been added on Page 8, Line 153-156.

  1. “sodium phosphate buffer” must be followed by abbreviation (PBS). See also other abbreviation used and explain them when, for the first time, appear in text: term (abbreviation)

Response:

Thank you very much for the constructive suggestions. We have added (PBS) after "sodium phosphate buffer". According to your suggestion, the description has been added on Page 9, Line 170.

  1. “… of Mojica et al (2015).” Reference must be included.

Response:

Thank you very much for the constructive suggestions. We have supplemented a reference in the revision manuscript.

Mojica, L., A. Meyer, M. A. Berhow, and E. G. de Mejia. "Bean Cultivars (Phaseolus Vulgaris L.) Have Similar High Antioxidant Capacity, in Vitro Inhibition of Alpha-Amylase and Alpha-Glucosidase While Diverse Phenolic Composition and Concentration." Food Research International 69 (2015): 38-48.

According to your suggestion, the description has been added on Page 48, Line 746-750.

  1. …. substrate pNPG (0.5 mol L-1 ) [20]. Reference 20 does not refer to this method. Mojica et al 2015 must appear in the References list instead of Zi-Cheng Zhang et al.

Response:

Thank you very much for the constructive suggestions. We have supplemented a reference in the revision manuscript and deleted the reference 20.

Mojica, L., A. Meyer, M. A. Berhow, and E. G. de Mejia. "Bean Cultivars (Phaseolus Vulgaris L.) Have Similar High Antioxidant Capacity, in Vitro Inhibition of Alpha-Amylase and Alpha-Glucosidase While Diverse Phenolic Composition and Concentration." Food Research International 69 (2015): 38-48.

According to your suggestion, the description has been added on Page 48, Line 746-750.

173-184. Too ambiguous, you must reformulate. It is understood that the control group was also injected with streptozocin. Also, you must specify that the metformin active substance is streptozocin.

Response:

Streptozotocin can be used to induce type 2 diabetes(T2DM). In the present study, mice in the blank control group did not require streptozotocin injection, while mice in other groups, including the metformin positive control group and the T2DM group, needed streptozotocin injection. Besides, metformin, as a positive drug, can be used as a control to compare the hypoglycemic effect of crocin I and geniposide.

200-201. “The mice's eyeballs were removed from the orbit for blood taking. Afterwards, the mice were executed by cervical dislocation”. The use of formulations like: “The retro-orbital site for blood sampling has been used; mice were euthanized by cervical dislocation” would be preferrable.

Response:

Thank you so much for your kind reminder. We replaced “The mice's eyeballs were removed from the orbit for blood taking. Afterwards, the mice were executed by cervical dislocation” with “The retro-orbital site for blood sampling has been used; mice were euthanized by cervical dislocation”. According to your suggestion, the description has been added on Page 12, Line 228-229.

  1. “should be recorded” replace with “were recorded”

Response:

Thank you so much for your kind reminder. We replaced “should be recorded” with “were record”. According to your suggestion, the description has been added on Page 13, Line 255.

  1. “single fragments of the compound’’. What do you mean by single fragment?

Figure 1A and 1B: Must be reunited in one figure and the title must be changed: Figure 1. HPLC chromatograms. A) λ=238 nm of crude EGJ (a), standard compound of geniposide (b), 20% EGJ (c) and No.4 sample (d). B: λ=440 nm of crude EGJ (a), standard compound of crocin I (b), No.8 sample (c), No.9 sample (d) and No.14 sample (e). Near all peaks should be indicated the retention time. The quality of figure 1B must be improved.

Response:

Fig. 1 HPLC chromatograms. A) λ=238 nm of crude EGJ (a), standard compound of geniposide (b), 20% EGJ (c) and No.4 sample (d). B: λ=440 nm of crude EGJ (a), standard compound of crocin I (b), No.8 sample (c), No.9 sample (d) and No.14 sample (e). According to your suggestion, the description has been added on Page 16, Line 276-279.

We indicated the retention time in the figure.

Figure 2 is the fragment of gardenia extract obtained by purification, inevitably there are some impurities, so the quality of the liquid phase diagram is not good, hoping to be understood

257-258. “No.8 and No.9 sample were at a concentration of 0.5 mg mL-1 and inhibition rates were 35.74 ± 4.86% and 97.61 ± 8.12%, respectively.” No 14 has a different concentration?

Response:

At the same concentration as samples No. 8 and No. 9,No.14 sample (21.79 ± 4.27%) had a slightly weaker inhibition ability than acarbose (35.99 ± 3.28%). According to your suggestion, the description has been added on Page 26, Line 387-388.

In Figure 2B change sample 8 and sample 9 with No. 8 and No. 9.

Response:

Thank you so much for your kind reminder. We replaced “sample 8 and sample 9” with “No. 8 and No. 9”.

  1. “According to "3.2" in this chapter” replace with “As mentioned before”

Response:

Thank you so much for your kind reminder. We replaced “in this chapter” with “as mentioned before”. According to your suggestion, the description has been added on Page 28, Line 412.

  1. “for the type of inhibition study’ replace with “to study the type of inhibition”

Response:

Thank you so much for your kind reminder. We replaced “for the type of inhibition” with “to study the type of inhibition”. According to your suggestion, the description has been added on Page 28, Line 413.

  1. ~ replace with –

Response:

Thank you so much for your kind reminder. We replaced “~” with “–”. According to your suggestion, the description has been added on Page 28, Line 415.

  1. Reference [23] does not refer to the aspects mentioned in the text. Replace it with a relevant one.

Response:

Thank you very much for the constructive suggestions. We have replaced this reference in the revision manuscript.

Yang, Yang, Jiu-liang Zhang, Lu-hong Shen, Lan-jie Feng, and Qing Zhou. "Inhibition Mechanism of Diacylated Anthocyanins from Purple Sweet Potato (Ipomoea Batatas L.) against Alpha-Amylase and Alpha-Glucosidase." Food Chemistry 359 (2021).

According to your suggestion, the description has been added on Page 28, Line 757-760.

  1. “According to "3.2.1" in this chapter” replace with “As stated before”

Response:

Thank you so much for your kind reminder. We replaced “in this chapter” with “as stated before”. According to your suggestion, the description has been added on Page 29, Line 426.

312-313. “After three days of STZ injection, the weight changes of the mice were shown in Fig. 5”. Move “…. After three days of STZ injection” after “mice”.

Response:

Thank you so much for your kind reminder. We replaced “After three days of STZ injection, the weight changes of the mice were shown in Fig. 5” with “The weight changes of the mice after three days of STZ injection were shown in Fig. 7”. According to your suggestion, the description has been added on Page 30, Line 445-446.

  1. “eighth” must be replaced with “eight”.

  1. “in this paper” replace with “in this study”

Response:

Thank you so much for your kind reminder. We replaced “in this paper” with “in this study”. According to your suggestion, the description has been added on Page 31, Line 469.

  1. “Figure 6. A: The changes of body weight. B: The changes water intake. C: The changes food intake.” Modify as “Figure 6. The changes of A: body weight. B: water intake. C: food intake.”

Response:

Thank you so much for your kind reminder. We modified “Figure 6. A: The changes of body weight. B: The changes water intake. C: The changes food intake.”  as “Figure 8. The changes of A: body weight. B: water intake. C: food intake.” According to your suggestion, the description has been added on Page 32, Line 480.

  1. “at 15 min” replace with “after 15 min”

Response:

Thank you so much for your kind reminder. We replaced “at 15 min” with “after 15 min”. According to your suggestion, the description has been added on Page 33, Line 495-496.

  1. “geniposide was better” add “fraction” to geniposide

Response:

Thank you so much for your kind reminder. We replaced “geniposide was better” with “geniposide fraction was better”. According to your suggestion, the description has been added on Page 35, Line 522.

Figure 9C. On the vertical axis, correct “HDH-L” with “HDL-C”

Response:

Thank you so much for your kind reminder. We replaced “HDH-L” with “HDL-C”.

Figure 10. You should align horizontally all 3 diagrams.

Response:

We have aligned horizontally all 3 diagrams in Figure 10.

Table2.

What is the significance of “-” before some values of ppm?

Response:

“-” indicates that the actual measured value is less than the theoretical value

H2O must be H2O

Response:

Thank you so much for your kind reminder. We replaced “H2O” with “H2O”.

Note 2:"+" and "-" refer to "deleted" and "not detected or content too low. Replace “deleted” with “detected”

Response:

Thank you so much for your kind reminder. We replaced “deleted” with “detected”. According to your suggestion, the description has been added on Page 25, Line 382.

Title of Figure 11. “A, B, C were TIC of geniposide and crocin I standards, crude EGJ, No.9 sample, respectively” must be improved

Response:

Fig. 2 Total Ion Chromatogram. (A) geniposide and crocin I standards, (B) crude EGJ, (C) No.9 sample

According to your suggestion, the description has been added on Page 19, Line 344-345.

Title of Figure 12. “A, B were TOF-MS and TOF-MS/MS spectra of geniposide, crocin I. As well as their mass spectrometric fragmentation pathway” must be improved.

Response:

Fig. 3 (A)TOF-MS, TOF-MS/MS spectra and mass spectrometric fragmentation pathway of geniposide. (B) TOF-MS, TOF-MS/MS spectra and mass spectrometric fragmentation pathway of crocin I.

According to your suggestion, the description has been added on Page 21, Line 377-379.

Section 5. Discussion.

100-103, 104-107. Affirmations must be sustained by literature references.

Response:

Thank you very much for the constructive suggestions. We have supplemented several references in the revision manuscript. Their details are as follows:

  1. Liu, Yujia, Jie Zhu, Jiamei Yu, Xu Chen, Shuyan Zhang, Yanxue Cai, and Lin Li. "A New Functionality Study of Vanillin as the Inhibitor for Alpha-Glucosidase and Its Inhibition Kinetic Mechanism." Food Chemistry 353 (2021).
  2. Tan, Sin Yee, Joyce Ling Mei Wong, Yan Jinn Sim, Su Sie Wong, Safa Abdelgadir Mohamed Elhassan, Sean Hong Tan, Grace Pei Ling Lim, Nicole Wuen Rong Tay, Naveenya Chetty Annan, Subrat Kumar Bhattamisra, and Mayuren Candasamy. "Type 1 and 2 Diabetes Mellitus: A Review on Current Treatment Approach and Gene Therapy as Potential Intervention." Diabetes & metabolic syndrome 13, no. 1 (2019): 364-72.
  3. Cakar, Uros, Nada Grozdanic, Boris Pejin, Vesna Vasic, Mira Cakar, Aleksandar Petrovic, and Brizita Djordjevic. "Impact of Vinification Procedure on Fruit Wine Inhibitory Activity against Alpha-Glucosidase." Food Bioscience 25 (2018): 1-7.
  4. Cakar, Uros, Nada Grozdanic, Aleksandar Petrovic, Boris Pejin, Branislav Nastasijevic, Bojan Markovic, and Brizita Dordevic. "Fruit Wines Inhibitory Activity against Alpha-Glucosidase." Current Pharmaceutical Biotechnology 18, no. 15 (2017): 1264-72.
  5. Ren, Shuncheng, Yi Wan, Linzheng Li, and Tianyi Pan. "Studies on the Inhibition Kinetics and Interaction Mechanism of Gardenia Yellow on Starch Digestive Enzyme." Journal of Chinese Institute of Food Science and Technology 21, no. 9 (2021): 38-47.

According to your suggestion, the description has been added on Page 49-50, Line 774-794.

134-138. “Some studies have found that gardeniside has hypoglycemic effect on diabetic model established by streptozotocin (STZ) in rats. The mechanism may be related to inhibiting the expression of NF-κB and Bax in apoptosis cells, inhibiting the activity of Caspase-3 and Caspase-9 protease, and enhancing the expression of Bcl-2 in apoptotic cells [29].” Reference [29] refers to geniposide, that stimulates glycogen synthesis in mice induced by a high-fat diet and streptozotocin injection, no mention about gardeniside. Correct the informations.

Response:

Thank you so much for your kind reminder. We have corrected this information.

“Studies suggested that the hypoglycemic effect of geniposide alleviated may be mediated by alleviating the glycogen phosphorylase and glucose-6-phosphatase activities”

According to your suggestion, the description has been added on Page 42, Line 619-621.

Reviewer 2 Report

I had a opportunity to review paper entitled Therapeutic effective verification of crocin I, geniposide and gardenia (Gardenia jasminoides Ellis) on type 2 diabetes mellitus in vivo-in vitro. This work is dealing with extract of gardenia in view of antidiabetic effect in in vivo-in vitro conditions. I think that this paper have a really good potential, but in some parts are very basic.

So, these are my suggestions for the improvment:

Basic suggestions:

Abstract is so generic, please revise using the Instructions for author. Rewrite the whole abstract and highlight in a better manner the aim of the study.

Think about revision of keywords.

All Equations must be numerated.

Figure 2 can be improved in presentation type and quality of graphs.

Figure 6 must have title, and after that legend explanation. The same comments for Figure 7.

The main suggestions:

Authors said that are less information about the functional mechanism of hypoglycemic effect of crocin I, geniposide from gardenia on T2DM in vivo-in vitro and highlight necessary to investigate and compare the intervention of geniposide, crocin I and extract of gardenia on hyperglycemic mice. But, I think that the Introduction must be expanded with more information about in vivo-in vitro experiment in this area with explanation of this type of study and significance of similar study. Also, why extract are significant, when and why they will be used. Emphasize differences between crude and pure extracts in this type of study.

Author Response

Thank you all so much for your reviewing my work (Manuscript ID:foods-2297227, Title: Therapeutic effective verification of crocin I, geniposide and gardenia (Gardenia jasminoides Ellis) on type 2 diabetes mellitus in vivo-in
vitro).

First, I want to show my respect to all the reviewers, because they are competent, and their comments and suggestions are highly reasonable, and help me a lot.

According to their comments, I have tried my best to revise my manuscript. And now, I will give my detailed explanations for the revision:

Editorial comments:

Reviewer 2

Basic suggestions:

Abstract is so generic, please revise using the Instructions for author. Rewrite the whole abstract and highlight in a better manner the aim of the study.

Response:

For many centuries, Gardenia (Gardenia jasminoides Ellis) is highly valued as a food homologous Chinese herbal medicine and various bioactive compounds are found in gardenia, of which crocin I and geniposide are representatives. However,the functional mechanism of underlying the hypoglycemic effect of gardenia is missing in literature. To evaluate the effect of gardenia and its different extracts on type 2 diabetes mellitus (T2DM) by in vivo-in vitro experiment, the dried powder of gardenia was extracted with 60% ethanol and eluted at different ethanol concentrations to obtain the corresponding purified fragments. After that, the active chemical composition of different purified fragments of gardenia was analyzed by HPLC. Then the hypoglycemic effects of different purified fragments of gardenia were compared by in vitro-in vivo experiments. Finally, the different extracts were characterized by UPLC-ESI-QTOF-MS/MS and the mass spectrometric fragmentation pathway of the two main compounds, geniposide and crocin I, were identified. Experimental results indicated that the inhibitory effect of 40% EGJ (crocin I) on α-glucosidase was better than that of 20% EGJ (geniposide) in vitro. However, the inhibitory effect of geniposide on T2DM was better than that of crocin I in the animal experiments. The different results in vivo-in vitro presumed that the possibly different mechanisms between crocin I and geniposide on T2DM. This research can demonstrate that the mechanism of hypoglycemia in vivo by geniposide is not only one target of α-glucosidase and provide experimental background for crocin I and geniposide deep processing and utilization.

According to your suggestion, the description has been added on Page 2, Line 22-42.

Think about revision of keywords.

Response:

Gardenia; α-glucosidase; crocin I; geniposide; type 2 diabetes mellitus

According to your suggestion, the description has been added on Page 3, Line 44-45.

All Equations must be numerated.

Response:

We numerated all equations.

Figure 2 can be improved in presentation type and quality of graphs.

Response:

Figure 2 is the fragment of gardenia extract obtained by purification, inevitably there are some impurities, so the quality of the liquid phase diagram is not good, hoping to be understood

Figure 6 must have title, and after that legend explanation. The same comments for Figure 7.

Response:

Fig. 8 The changes of A: body weight. B: water intake. C: food intake.

Fig. 9 A, B and C represent the changes of FBG, OGTT and AUC, respectively. Note: Data were expressed as the mean ± SD (n = 9) and different letters marked above the bars showed significant differences (p < 0.05).

According to your suggestion, the description has been added on Page 32, Line 480 and Page 34, Line 510-512.

The main suggestions:

Authors said that are less information about the functional mechanism of hypoglycemic effect of crocin I, geniposide from gardenia on T2DM in vivo-in vitro and highlight necessary to investigate and compare the intervention of geniposide, crocin I and extract of gardenia on hyperglycemic mice. But, I think that the Introduction must be expanded with more information about in vivo-in vitro experiment in this area with explanation of this type of study and significance of similar study. Also, why extract are significant, when and why they will be used. Emphasize differences between crude and pure extracts in this type of study.

Response:

Crocin I accounts for 70% of crocetin[14]. As crocetin is rare and precious, it has become the main trend to get it from gardenia. In recent years, it has been found to have many pharmacological activities, such as anti-inflammatory, tumor prevention, cardiovascular disease prevention and as water-soluble nutrients can be easily absorbed in the body[15-17].

According to your suggestion, the description has been added on Page 6, Line 90-94.

One study showed that show gardenia latifolia extract (GLE) has antidiabetic potential in rats with high-fat diet (HFD) + streptozotocin (STZ) induced type 2 diabetes mellitus (T2DM)[18].

According to your suggestion, the description has been added on Page 6, Line 94-96.

Reviewer 3

Comments:

  1. Line 72- „The objective of present study was to compare and identify the structure of bioactive compounds of gardenia in vivo and vitro experiments“. This sentence is not entirely clear. Please reconstruct it.

Response:

The objective of the present study was to compare the hypoglycemic effects of bioactive compounds from gardenia in in vivo and in vitro experiments and to identify the structural of them. According to your suggestion, the description has been added on Page 6, Line 101-103.

  1. Line 103- „According to the results of preliminary experiments, the elution effect of AB-8 macroporous adsorption resin was the best“. If there is, provide a reference or clarify what kind of preliminary experiments are involved.

Response:

Zhang, Jiu-Liang, Chun-Li Luo, Qing Zhou, and Zi-Cheng Zhang. "Isolation and Identification of Two Major Acylated Anthocyanins from Purple Sweet Potato (Ipomoea Batatas L. Cultivar Eshu No. 8) by Uplc-Qtof-Ms/Ms and Nmr." International Journal of Food Science and Technology 53, no. 8 (2018): 1932-41.

According to your suggestion, the description has been added on Page 47, Line 732-736.

  1. Line 121- „Sample No.9 showed the best inhibition of α-glucosidase“. There is no need to write this in the Materials and methods part.

Response:

Thank you very much for the constructive suggestions. We deleted this fragment in the revised manuscript and this fragment was moved in section 3.2.

  1. It is necessary to clarify the abbreviations in Scheme 1 and Figure 6. Also, it is necessary to flip the numbers on the x-axis in Figure 6.

Response:

Thank you very much for the constructive suggestions. The abbreviations in Scheme 1 and Figure 6 are shown in the abbreviation section at the beginning of the article. And we have flipped the numbers on the x-axis in Figure 6.

  1. Pay attention to writing in vivo and in vitro. It should be italic (line 468, 470, 156, 157).

Response:

Thank you very much for the constructive suggestions. We have adjusted in vivo and in vitro to italic form.

  1. Please clarify how did you identified compounds using UPLC-ESI-QTOF-MS/MS?

Response:

According to the primary and secondary mass spectra of geniposide, it is known that the mass-to-charge ratio (m/z) of geniposide with sodium ([M+Na]+) is 411.1267, 777.2850 corresponds to two geniposide couples ([2M+H]+) , 389.1268 is geniposide ([M+H]+); fragment ion 227.0921 is geniposide stripped of one molecule of monosaccharide ([M-glc+H]+), 209.0805 for geniposide stripped of one molecule of monosaccharide and one molecule of water ([M-glc-H2O+H]+), as shown in Fig. 3 A. It is presumed that geniposide is relatively easy to remove one molecule of monosaccharide, followed by further removal of one molecule of water from the parent nucleus of cyclic enol ether terpene, and its mass spectrometry cleavage pathway is shown in Fig. 3 A:

On the basis of the primary and secondary mass spectra of crocin I, the mass-to-charge ratio (m/z) of crocin I ([M+Na]+) with sodium ions is 999.3741, 976.3801 corresponds to crocin I ([M+H]+); fragment ions 675.2639 are crocin I with sodium ions stripped of disaccharides ([M-2glc+Na]+), 329.1755 for crocin I stripped of two disaccharides ([M-4glc]+), and 347.0957 for one molecule of disaccharide with sodium ions ([M-2glc-C20H20O3]+), as shown in Fig. 3 B. It is presumed that the disaccharide at both ends of crocin I is relatively easy to break the glycosidic bond attached to the parent chain, and its mass spectral cleavage pathway is shown in Fig. 3 B:

According to your suggestion, the description has been added on Page 17, Line 282-310.

Reviewer 3 Report

In this paper, the authors tried to evaluate the effect of gardenia and its different extracts on type 2 diabetes mellitus by in vivo/in vitro experiments. The authors did an extensive work, but there are some comments and questions that need to be addressed.

Comments:

1. Line 72- „The objective of present study was to compare and identify the structure of bioactive compounds of gardenia in vivo and vitro experiments“. This sentence is not entirely clear. Please reconstruct it.

2. Line 103- „According to the results of preliminary experiments, the elution effect of AB-8 macroporous adsorption resin was the best“. If there is, provide a reference or clarify what kind of preliminary experiments are involved.

3. Line 121- „Sample No.9 showed the best inhibition of α-glucosidase“. There is no need to write this in the Materials and methods part.

4. It is necessary to clarify the abbreviations in Scheme 1 and Figure 6. Also, it is necessary to flip the numbers on the x-axis in Figure 6.

5. Pay attention to writing in vivo and in vitro. It should be italic (line 468, 470, 156, 157).

6. Please clarify how did you identified compounds using UPLC-ESI-QTOF-MS/MS?

Author Response

Thank you all so much for your reviewing my work (Manuscript ID:foods-2297227, Title: Therapeutic effective verification of crocin I, geniposide and gardenia (Gardenia jasminoides Ellis) on type 2 diabetes mellitus in vivo-in
vitro).

First, I want to show my respect to all the reviewers, because they are competent, and their comments and suggestions are highly reasonable, and help me a lot.

According to their comments, I have tried my best to revise my manuscript. And now, I will give my detailed explanations for the revision:

Editorial comments:

Reviewer 3

Comments:

  1. Line 72- „The objective of present study was to compare and identify the structure of bioactive compounds of gardenia in vivo and vitro experiments“. This sentence is not entirely clear. Please reconstruct it.

Response:

The objective of the present study was to compare the hypoglycemic effects of bioactive compounds from gardenia in in vivo and in vitro experiments and to identify the structural of them. According to your suggestion, the description has been added on Page 6, Line 101-103.

  1. Line 103- „According to the results of preliminary experiments, the elution effect of AB-8 macroporous adsorption resin was the best“. If there is, provide a reference or clarify what kind of preliminary experiments are involved.

Response:

Zhang, Jiu-Liang, Chun-Li Luo, Qing Zhou, and Zi-Cheng Zhang. "Isolation and Identification of Two Major Acylated Anthocyanins from Purple Sweet Potato (Ipomoea Batatas L. Cultivar Eshu No. 8) by Uplc-Qtof-Ms/Ms and Nmr." International Journal of Food Science and Technology 53, no. 8 (2018): 1932-41.

According to your suggestion, the description has been added on Page 47, Line 732-736.

  1. Line 121- „Sample No.9 showed the best inhibition of α-glucosidase“. There is no need to write this in the Materials and methods part.

Response:

Thank you very much for the constructive suggestions. We deleted this fragment in the revised manuscript and this fragment was moved in section 3.2.

  1. It is necessary to clarify the abbreviations in Scheme 1 and Figure 6. Also, it is necessary to flip the numbers on the x-axis in Figure 6.

Response:

Thank you very much for the constructive suggestions. The abbreviations in Scheme 1 and Figure 6 are shown in the abbreviation section at the beginning of the article. And we have flipped the numbers on the x-axis in Figure 6.

  1. Pay attention to writing in vivo and in vitro. It should be italic (line 468, 470, 156, 157).

Response:

Thank you very much for the constructive suggestions. We have adjusted in vivo and in vitro to italic form.

  1. Please clarify how did you identified compounds using UPLC-ESI-QTOF-MS/MS?

Response:

According to the primary and secondary mass spectra of geniposide, it is known that the mass-to-charge ratio (m/z) of geniposide with sodium ([M+Na]+) is 411.1267, 777.2850 corresponds to two geniposide couples ([2M+H]+) , 389.1268 is geniposide ([M+H]+); fragment ion 227.0921 is geniposide stripped of one molecule of monosaccharide ([M-glc+H]+), 209.0805 for geniposide stripped of one molecule of monosaccharide and one molecule of water ([M-glc-H2O+H]+), as shown in Fig. 3 A. It is presumed that geniposide is relatively easy to remove one molecule of monosaccharide, followed by further removal of one molecule of water from the parent nucleus of cyclic enol ether terpene, and its mass spectrometry cleavage pathway is shown in Fig. 3 A:

On the basis of the primary and secondary mass spectra of crocin I, the mass-to-charge ratio (m/z) of crocin I ([M+Na]+) with sodium ions is 999.3741, 976.3801 corresponds to crocin I ([M+H]+); fragment ions 675.2639 are crocin I with sodium ions stripped of disaccharides ([M-2glc+Na]+), 329.1755 for crocin I stripped of two disaccharides ([M-4glc]+), and 347.0957 for one molecule of disaccharide with sodium ions ([M-2glc-C20H20O3]+), as shown in Fig. 3 B. It is presumed that the disaccharide at both ends of crocin I is relatively easy to break the glycosidic bond attached to the parent chain, and its mass spectral cleavage pathway is shown in Fig. 3 B:

According to your suggestion, the description has been added on Page 17, Line 282-310.

Round 2

Reviewer 1 Report

·       Choose between “Gardenia” and “gardenia”

·       Use normal characters for comma in the whole manuscript (line 395, 396, etc.)

·       Between word and comma no space needed

·       Between text and “[“ add space  

·       Between “.“ and text add space (line 113, 128, etc.)

·       Line 95. One study showed that show gardenia latifolia extract (GLE) has antidiabetic potential in rats with high-fat

·       Line 103. the structural of them.

·       Line 105. And different concentrations

·       Line 108-112. This paper explored that the hypoglycemic effect of geniposide, crocin I and EGJ intervention role playing on T2DM and it can also enrich the potential use of gardenia as a hypoglycemic functional food and lay the foundation for new high added-value product development of gardenia.

·       Line 133-134. Add 1:7 volume ratio of petroleum ether to the viscous infusion, and acquire the crude extract of gardenia (EGJ) after concentrating and vacuum freeze-drying. was added, was acquired.

·       Line 148. solvent B (Methanol)

·       Line 193. During in the study period

·       You must specify that the metformin active substance is streptozocin.

·       Line 210. The mice in each group were administrated intragastric for 5 weeks. Explain what was administrated to mice, or change the phrase. …. were fed…?

·       The two figures 1A and 1B should be presented as a single figure or named as figure 1 and 2

·       Line 302. shown in Fig. 3A:  remove “:” Do the same for line 310

·       Fig. 3. The fonts in fragmentation pathway are different, please use the same font. The title must be changed, as for example: Fig. 3.  TOF-MS, TOF-MS/MS spectra and mass spectrometric fragmentation pathway of (A) geniposide and (B) crocin I.

·       Line 381. Note 1: glc glucose; add “,” between glc and glucose

·       Line 431. the size of the intercept and the position of the axes. The position of axes is fixed. The coordinates are changed. Correct accordingly

·       Line 431. The results are shown in Fig. 6: an…  delete the “:” and make two sentences

·       Line 458. degrees. the body weight…  The

·       Line 460. that the weight of T2DM mice could be maintained by metformin….. controlled

Author Response

Thank you all so much for your reviewing my work (Manuscript IDfoods-2297227, Title: Therapeutic effective verification of crocin I, geniposide and gardenia (Gardenia jasminoides Ellis) on type 2 diabetes mellitus in vivo-in
vitro
).

First, I want to show my respect to all the reviewers, because they are competent, and their comments and suggestions are highly reasonable, and help me a lot.

According to their comments, I have tried my best to revise my manuscript. And now, I will give my detailed explanations for the revision:

Editorial comments:

Reviewer 1

Comments and Suggestions for Authors

Choose between “Gardenia” and “gardenia”

Response:

I chose “gardenia” because it fits this article better

Use normal characters for comma in the whole manuscript (line 395, 396, etc.)

Response:

We use normal characters for comma in the whole manuscript. According to your suggestion, the description has been added on Page 26, Line 395,396.

Between word and comma no space needed

Response:

We removed the space between word and comma.

Between text and “[“ add space  

Response:

We added space between text and “[”.

Between “.“ and text add space (line 113, 128, etc.)

Response:

We added the space between “.” and text. According to your suggestion, the description has been added on Page 7, Line. 113,128, etc.

Line 95. One study showed that show gardenia latifolia extract (GLE) has antidiabetic potential in rats with high-fat

Response:

One study showed that gardenia latifolia extract (GLE) has antidiabetic potential in rats with high-fat.

We deleted the “show”. According to your suggestion, the description has been added on Page 6, Line. 94-95.

 Line 103. the structural of them.

Response:

the structure of them.

We replaced the “structural” with “structure”. According to your suggestion, the description has been added on Page 6, Line. 102.

Line 105. And different concentrations

Response:

Then the different concentrations

We replace “and” with “then”. According to your suggestion, the description has been added on Page 6, Line. 104-105.

Line 108-112. This paper explored that the hypoglycemic effect of geniposide, crocin I and EGJ intervention role playing on T2DM and it can also enrich the potential use of gardenia as a hypoglycemic functional food and lay the foundation for new high added-value product development of gardenia.

Response:

This paper explored the hypoglycemic effect of geniposide, crocin I and EGJ intervention role playing on T2DM and this research can also enrich the potential use of gardenia as a hypoglycemic functional food and lay the foundation for new high added-value product development of gardenia.

We deleted the “that” and replace “it” with “this research”. According to your suggestion, the description has been added on Page 6, Line.106-110

Line 133-134. Add 1:7 volume ratio of petroleum ether to the viscous infusion, and acquire the crude extract of gardenia (EGJ) after concentrating and vacuum freeze-drying. was added, was acquired.

Response:

The 1:7 volume ratio of petroleum ether was added to the viscous infusion, and the crude extract of gardenia (EGJ) was acquired after concentrating and vacuum freeze-drying.

According to your suggestion, the description has been added on Page 7, Line.131-133.

Line 148. solvent B (Methanol)

Response:

We replaced “Methanol” with “methanol”. According to your suggestion, the description has been added on Page 8, Line.147.

Line 193. During in the study period

Response:

During the experimental period

According to your suggestion, the description has been added on Page 10, Line.192.

You must specify that the metformin active substance is streptozocin.

Response:

Dear reviewer, Thank you so much for your reviewing my work. I'm sorry, maybe I'm not very clear about the description of animal experiments

Metformin and streptozocin are two very different substances. In this experiment, Kunming mice were used as the study object, and streptozocin was taken for intraperitoneal injection to cause hyperglycemic symptoms by damaging the pancreatic β cells of mice, while the mice were fed with high-fat diet for a long time to simulate the diet of diabetic patients. But metformin is used for the first-line clinical treatment for hyperglycemia in patients with type 2 diabetes, which was used as a positive control group in this research to compare the hypoglycemic effect of crocin Ⅰ and geniposide.

The animal type 2 diabetes model was established as follows:

According to the situation of adaptive feeding and the weight of mice, 9 mice were chosen as the normal control group and continued to be fed with ordinary diet. Other mice were fed with high-fat diet. Each group consisted of 9 mice. Four weeks later, the mice of high-fat diet group were intraperitoneally injected with streptozotocin (STZ, 65 mg kg-1bw-1) for 3 consecutive days. Then, the mice were kept feeding with high-fat diet for one week. Fasting blood glucose of the mice were measured after fasting for 12 hours. The group with fasting blood glucose value ≥ 7.8 mmol/L could be regarded as the successful T2DM mice.

Line 210. The mice in each group were administrated intragastric for 5 weeks. Explain what was administrated to mice, or change the phrase. …. were fed…?

The mice in each group were fed for 5 weeks.

Response:

According to your suggestion, the description has been added on Page 11, Line.209.

The two figures 1A and 1B should be presented as a single figure or named as figure 1 and 2

Response:

The two figures 1A and 1B were presented as figure 1 and 2, respectively.

Line 302. shown in Fig. 3A:  remove “:” Do the same for line 310

Response:

We replaced “:”with “.”. According to your suggestion, the description has been added on Page 17, Line.298 and Page 18, Line.306.

Fig. 3. The fonts in fragmentation pathway are different, please use the same font. The title must be changed, as for example: Fig. 3.  TOF-MS, TOF-MS/MS spectra and mass spectrometric fragmentation pathway of (A) geniposide and (B) crocin I.

Response:

We used the same font in fragmentation pathway and the title was be change as Fig. 4.  TOF-MS, TOF-MS/MS spectra and mass spectrometric fragmentation pathway of (A) geniposide and (B) crocin I.

According to your suggestion, the description has been added on Page 17, Line.345-346.

Line 381. Note 1: glc glucose; add “,” between glc and glucose

Response:

We added “,” between glc and glucose. According to your suggestion, the description has been added on Page 25, Line.348.

Line 431. the size of the intercept and the position of the axes. The position of axes is fixed. The coordinates are changed. Correct accordingly

Response:

which was judged by the size of the intercept and the position of the coordinate

we replaced “axes” with “coordinate”. According to your suggestion, the description has been added on Page 29, Line.398.

Line 431. The results are shown in Fig. 6: an…  delete the “:” and make two sentences

Response:

The results are shown in Fig. 7, when the sample concentration increases, the vertical coordinate of the corresponding line grows, indicating that 1/V increases and Vmax decreases. While a decrease in the horizontal coordinate indicates a decrease in -1/Km and an increase in the Mie's constant Km.

We deleted the “:” and made two sentences. According to your suggestion, the description has been added on Page 29, Line.398-402.

Line 458. degrees. the body weight…  “The

Response:

We replaced “the” with “The”. According to your suggestion, the description has been added on Page 30, Line.425.

Line 460. that the weight of T2DM mice could be maintained by metformin….. controlled

Response:

We replaced “maintained” with “controlled”. According to your suggestion, the description has been added on Page 30, Line.427.

Reviewer 2 Report

/

Author Response

Thank you so much for your reviewing my work (Manuscript IDfoods-2297227, Title: Therapeutic effective verification of crocin I, geniposide and gardenia (Gardenia jasminoides Ellis) on type 2 diabetes mellitus in vivo-in
vitro
).

First, I want to show my respect to all the reviewers, because they are competent, and their comments and suggestions are highly reasonable, and help me a lot.

According to their comments, I have tried my best to revise my manuscript. And now, I will give my detailed explanations for the revision:

Editorial comments:

Reviewer 2

Thanks for all your careful comments and suggestions.

Reviewer 3 Report

Thank you for answering my questions. In my opinion, the paper is now suitable for publication.

Author Response

Thank you so much for your reviewing my work (Manuscript IDfoods-2297227, Title: Therapeutic effective verification of crocin I, geniposide and gardenia (Gardenia jasminoides Ellis) on type 2 diabetes mellitus in vivo-in
vitro
).

First, I want to show my respect to all the reviewers, because they are competent, and their comments and suggestions are highly reasonable, and help me a lot.

According to their comments, I have tried my best to revise my manuscript. And now, I will give my detailed explanations for the revision:

Editorial comments:

Reviewer 3

Thanks for all your careful comments and suggestions.